# Practical Lessons on Vector-Symbolic Architectures in Deep Learning-Inspired Environments

**Francesco Carzaniga** FRC@ZURICH.IBM.COM
*IBM Research – Zurich and Department of Neurology, Inselspital, Sleep-Wake-Epilepsy-Center, Bern University Hospital, Bern University, Bern, Switzerland*

**Michael Hersche**
*IBM Research – Zurich*

**Kaspar Schindler**
*Department of Neurology, Inselspital, Sleep-Wake-Epilepsy-Center, Bern University Hospital, Bern University, Bern, Switzerland*

**Abbas Rahimi**
*IBM Research – Zurich*

**Editors:** Leilani H. Gilpin, Eleonora Giunchiglia, Pascal Hitzler, and Emile van Krieken

## Abstract

Neural networks have shown unprecedented capabilities, rivaling human performance in many tasks. However, current neural architectures are not capable of symbolic manipulation, which is thought to be a hallmark of human intelligence. Vector-symbolic architectures (VSAs) promise to bring this ability through simple vector manipulation, highly amenable to current and emerging hardware and software stacks built for their neural counterparts. Integrating the two models into the paradigm of neuro-vector-symbolic architectures may achieve even more human-like performance. However, despite ongoing efforts, there are no clear guidelines on the deployment of VSA in deep learning-based training situations. In this work, we aim to begin providing such guidelines by offering four practical lessons we have observed through the analysis of many VSA models and implementations. We provide thorough benchmarks and results that corroborate such lessons. First, we observe that Multiply-add-permute (MAP) and Hadamard linear binding (HLB) are up to 3–4× faster than holographic reduced representations (HRR), even when the latter is equipped with optimized FFT-based convolutions. Second, we propose further speed improvements by replacing similarity search with a linear readout, with no effect on retrieval. Third, we analyze the retrieval performance of MAP, HRR and HLB in a noise-free and noisy scenario to simulate processing by a neural network, and show that they are equivalent. Finally, we implement a hierarchical multi-level composition scheme, with notable benefits to the flexibility of integration of VSAs inside existing neural architectures. Overall, we show that these four lessons lead to faster and more effective deployment of VSA.

## 1. Introduction

Symbolic manipulation is one of the hallmark characteristics of the brain (17; 30), and is thought to be a significant contributor to our ability to reason about the world. While deep neural networks (DNNs) have shown outstanding performance in many of the tasks commonly associated with human-level reasoning (33; 18; 46), they suffer from a distinct lack of symbolic manipulation reflected in poor out-of-distribution generalization (29; 5). Vector-symbolic architectures (VSAs) are a promising approach to perform symbolic manipulation

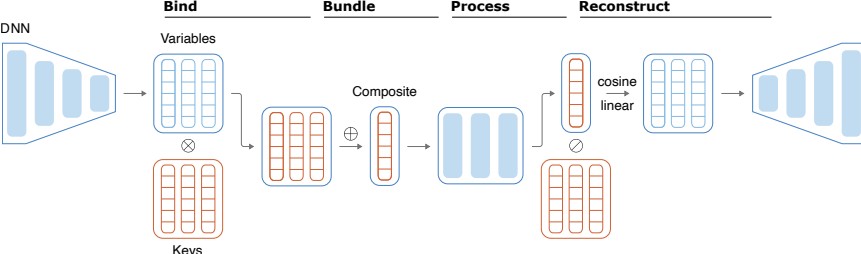

Figure 1: Typical implementation of VSA composition inside a neural network. Multiple internal representations are composed into one using binding ($\otimes$) and bundling (+). The composite representation can then further be processed, e.g., using VSA operators or a neural network, before it is reconstructed using unbinding ($\oslash$) and similarity searches. In blue are the neural components of the architecture, while in red are the symbolic components.

by means of mathematical operations on large $d$-dimensional vectors (10; 43; 34; 19). As simple mathematical operations on vectors are the bread and butter of DNNs as well, VSAs can be seamlessly integrated into DNNs and take advantage of the same hardware and software platforms (24; 44), leading to performant and efficient development (21).

VSAs can be deployed as memory analogues to increase the ability of DNNs (27) to acquire and retain information, during few-shot learning (22) and even zero-shot learning (38). Moreover, VSAs can enable class-incremental learning (15) with minimal computational expense (20). Inside Transformers (47) VSAs have been used to either replace (3) or enhance self-attention (40). VSAs can also be used inside DNNs to simultaneously process multiple image or text inputs (32) or efficiently encode time information (42). VSAs can effectively tackle extreme classification problems (9; 4), and hybrid architectures based on resonator networks (7; 23; 28; 37; 36) offer even stronger benefits for performing disentanglement. Finally, VSAs can be used to implement symbolic reasoning in hybrid, neuro-vector-symbolic architectures (NVSAs). Such models promise to combine DNNs and VSAs to achieve both high-level raw performance and human-like reasoning capabilities (39). Many NVSAs have achieved super-human performance in common intelligence benchmarks (16; 45).

Therefore, it is crucial to evaluate the compatibility of VSAs with DNN-based training practices. On this front, recent progress has showcased benefits to overall performance and computational complexity (12; 4). Indeed, VSAs are a family of methods with different characteristics and advantages, and it can be unclear which member of the family is best for any given task (19; 41; 25). Some VSA models, for example tensor product representations (TPR (43)), suffer from $O(d^2)$ memory scaling, making them unsuitable for DNNs (40). In contrast, other models such as holographic reduced representations (HRR (34; 35)), Fourier HRR (FHRR (34; 35)) and vector-derived transformation binding (VTB (12)) present a constant memory usage, together with quadratic or log-linear time complexity. Even faster, with a linear runtime, multiple add permute (MAP (11)) and Hadamard linear binding (HLB (4)) appear as the best candidates for DNN integration. While recent developments (48; 8; 4) push the boundary for efficient computation, popular

libraries such as TorchHD (13) and NengoSPA (1) have not necessarily been optimized for GPUs, the main accelerators for DNNs.

We aim to provide practical guidelines for the deployment of VSA families in a DNN-inspired setting, in terms of their runtime efficiency, their memory efficiency, and their retrieval accuracy. We focus here on the composition and retrieval task (Figure 1). Composition compresses internal DNN states into compact vectors, while retrieval aims to recover this information accurately. We choose three VSA families to evaluate: (1) HRR, the most popular model with quadratic or log-linear complexity; (2) MAP, an effective linear alternative; (3) HLB, a modern development again with linear complexity.

Overall, in this work, we showcase four practical lessons that can be followed to enhance the interface between VSAs and DNNs, leading to better NVSAs: (1) Prefer linear VSAs or GPU-optimized implementations of HRR; (2) Replace cosine similarity with a linear readout; (3) Use linear VSAs or HRR equivalently under noise-free or noisy conditions; (4) Use weighted multi-level compositions to create hierarchical data structures.

Following these four conclusions would lead to an improved usage of the existing computational resources and facilitate the onboarding of VSA into deep learning pipelines, to the benefit of the the field at large. Recent NVSA reasoning models, especially, can directly benefit from these four lessons. In the exemplary case of (45), replacing HRR with HLB both in the front-end and back-end could lead to faster training and inference, encouraging the development of bigger models. Moreover, the same attribute retrieval that leverages cosine similarity could be achieved through a linear readout for considerable speed and memory benefits.

Combining all four lessons above, we improve the integration of VSAs into DNNs without risk of computational bottlenecks and unwieldy representations.

## 2. Background

VSAs are a computational framework for encoding, manipulating, and reasoning about symbols in high-dimensional spaces (41; 25). In VSA, all symbols are $d$-dimensional vectors and can be manipulated using simple mathematical operations (19; 34). The vectors are drawn from a given vector space and combined using binding and bundling operations. Using these vectors and their operations, it is possible to both build and decompose rich representational structures in a way that is efficient even with common hardware. Finally, these representations are distributed, making them particularly interesting for neuromorphic hardware, which complement traditional DNNs while consuming orders of magnitude less energy (21; 28; 37). We now focus on the characteristic operations of VSA.

**Binding.** Binding is any function $\otimes : \mathbb{R}^d \times \mathbb{R}^d \to \mathbb{R}^k$ that composes two symbols (or vectors) into a new vector that is dissimilar to both constituent vectors. The two vectors can be interpreted as the key and the variable: the key is generated a priori and allows us to retrieve the variable, which is the symbol of interest, through unbinding ($\oslash$). Binding is often the bottleneck both in terms of computational complexity and memory requirements.

**Bundling.** Bundling is any function $+ : \mathbb{R}^d \times \mathbb{R}^d \to \mathbb{R}^d$ that composes two vectors into a new vector that is similar to both constituents. Bundling is used to represent a set of symbols. In this work, we only use elementwise addition, which has been shown to be fast and effective in most use cases.

**Composition.** We combine binding and bundling to create sets of symbols that can also serve to compress multiple vectors into one. This is particularly useful when VSA is used in conjunction with DNNs, as it enables the compression of multiple internal representations into one in a reversible manner. The composition and approximate decomposition of vectors is the target of this work.

**Similarity.** A similarity function sim $: \mathbb{R}^d \times \mathbb{R}^d \to \mathbb{R}$ computes the degree of relatedness between two vectors. As we have seen previously, an effective binding operation produces vectors that are unrelated (or dissimilar) to its inputs, while an effective bundling operation produces vectors that are related (or similar) to its inputs. Therefore, a good choice of similarity function is an important element in a VSA model. The typical choice is the cosine similarity $\sigma(\boldsymbol{x}, \boldsymbol{y}) = \dfrac{\boldsymbol{x} \cdot \boldsymbol{y}}{|\boldsymbol{x}||\boldsymbol{y}|}$.

**Similarity search.** The similarity function is also used to perform a search or cleanup. In particular, given a composite vector, we want to retrieve the best matching vector out of all the symbols we have instantiated. As both the dimension $d$ of the vectors and the size $L$ of the dictionary increase in sizes, so do the computation and memory complexity of performing this search. In the case of cosine similarity, a naïve implementation (such as the one present in PyTorch) requires materializing an intermediate matrix of $d \times d \times L$ elements, making such a search unfeasible.

## 2.1. VSA models

To properly introduce VSAs for the purpose of this work, we describe in more detail the two models we will take into consideration, with their advantages and drawbacks.

**Holographic reduced representations (HRR ([34]; [35])).** HRR is a popular choice of VSA, especially for integration with DNNs. Binding is performed by circular convolution, and unbinding is performed by approximating deconvolution via circular convolution with the inverse of the key (or, equivalently, via circular correlation). Hence, naïvely implementing HRR incurs in a $O(d^2)$ cost both for binding and unbinding. However, the cost can be lowered to $O(d \log d)$ using fast Fourier transform (FFT). Finally, bundling is done by elementwise addition. The memory capacity of HRR has been well characterized ([41]), making it often the best choice for many applications.

**Multiply add permute (MAP ([11])).** MAP is often chosen as a viable alternative to HRR, with the advantage of linear-time binding and unbinding operations in the form of Hadamard products. There are three major variants of MAP, namely MAP-I, MAP-B, and MAP-C. They are distinguished by their base distribution, which then also informs the bundling operation. In particular, MAP-B requires performing a *thresholding* operation, where resulting entries become either $1$ or $-1$ based on a threshold (usually set at half the bundled vectors). MAP-C instead requires *cutting* result entries at $1$ and $-1$ if they become greater (or smaller) than that limit. Finally, MAP-I uses a simpler sum.

**Hadamard linear binding (HLB ([4])).** HLB is a recent development in the field of VSA, offering a linear-time approach to both binding and unbinding analogous to MAP. Specifically, HLB can be seen as a real continuation of MAP-I, with some theoretical benefits in terms of computation stability.

Table 1 summarizes the main properties of the two VSA models considered here.

Table 1: Summary of the properties of the VSA families considered. $\mathcal{N}$ is the normal distribution, while $\mathcal{B}$ is the Bernoulli distribution.

| | Space | Distribution | Binding ($\otimes$) | Unbinding ($\oslash$) | Bundling ($+$) | Binding complexity |
|---|---|---|---|---|---|---|
| HRR (34; 35) | $\mathbb{R}^d$ | $\mathcal{N}(0, \frac{1}{d})$ | Circular convolution | Circular correlation | Sum | $O(d^2)\|O(d\log d)$ |
| HLB (4) | $\mathbb{R}^d$ | $\mathcal{N}(-\mu, \frac{1}{d}), \mathcal{N}(\mu, \frac{1}{d})$ | Hadamard product | Hadamard product | Sum | $O(d)$ |
| MAP-B (11) | $\{1, -1\}^d$ | $\mathcal{B}(0.5) \times 2 - 1$ | Hadamard product | Hadamard product | Sum (w/ threshold) | $O(d)$ |
| MAP-C (11) | $\mathbb{R}^d$ | $\mathcal{U}(-1, 1)$ | Hadamard product | Hadamard product | Sum (w/ cutting) | $O(d)$ |
| MAP-I (11) | $\mathbb{Z}^d$ | $\mathcal{B}(0.5) \times 2 - 1$ | Hadamard product | Hadamard product | Sum | $O(d)$ |

## 3. Methods

We aim to evaluate the performance of multiple VSA families in deep-learning inspired pipelines. As such, we choose tasks that are representative to a common usage of VSA in such pipelines. On the one hand, we evaluate the runtime and memory complexity of VSA on GPUs in multiple scenario, to distinguish between performant and less-performant variants and implementations. On the other hand, we evaluate the trade-offs between speed and accuracy, where they occur and whether there is need of mitigation strategies.

We now define the main task used in this work, composition. Let $d$ be the embedding size and $N$ the number of bundled vectors, first we create two codebooks $K$ and $V$ both with $1.5 \times d$ vectors following each VSA recipe. This enforces the pseudo-orthogonality of all vectors. Next, we randomly generate $N$ indices $I_{i \in [1, ..., N]}$, $I_i \leq 2 \times N$ and perform the composition

$$\boldsymbol{c} = \sum_{i=1}^{N} K_i \otimes V_{I_i} \tag{1}$$

$\boldsymbol{c}$ is a $d$-dimensional composition vector which contains key-value pairs corresponding to our indices. To recover all $I_i$, we perform the decomposition and similarity search

$$\hat{I}_i = \operatorname{argmax}_i(\operatorname{sim}(\boldsymbol{c} \oslash K, V)) \tag{2}$$

Finally, the accuracy is computed between $I$ and $\hat{I}$.

We vary the embedding size $d$ from 256 to 4096 in steps of 32, with multiples of 256 being most optimal for Nvidia GPUs (2). We also vary $N$ from 8 to 128 in steps of 8, to vary the task hardness. For each size, we perform multiple runs in parallel by using a batch size of 32: this produces statistically robust results. For runtime analysis, we fix $N = 128$ and repeat each run 15 times, discarding the top and bottom 20% to account for GPU warmup and caching. and report the average. Overall, for each embedding size we bind, bundle, and unbind 61440 pairs of vectors.

Now we define the four task variants.

**Reduced precision.** Given the prevalence of reduced precision computations in DNN training, we test the reconstruction accuracy of all VSA families using bf16 reduced precision. All other settings remain identical.

**Linear readout.** We replace the similarity search with a linear readout as follows

$$\hat{I}_i = \operatorname{argmax}_i((\boldsymbol{c} \oslash K) \times f(V)) \tag{3}$$

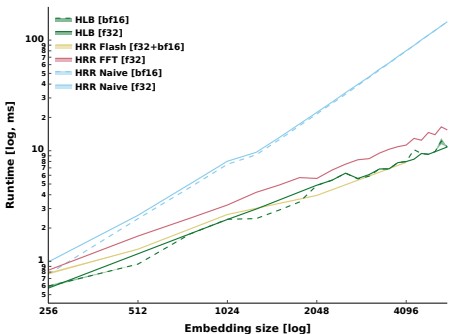
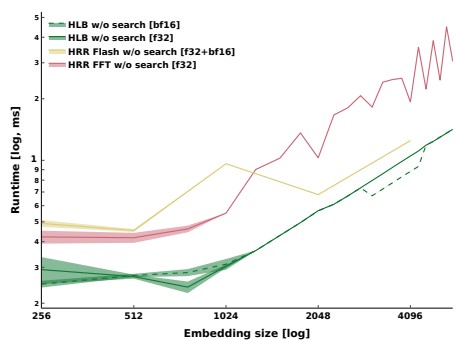

($a$) Similarity search is performed using cosine similarity.

($b$) A linear readout is used to bypass the similarity search.

Figure 2: Runtime comparison for composition and retrieval between different implementations of HRR, and HLB.

with $f$ either transpose or pseudoinverse. Note that more optimal readouts exist, such as the MMSE estimator (6), but they incur additional costs in an increased memory consumption or in the requirement of a calibration dataset to estimate the readout matrix. Therefore, we do not evaluate them in this work.

**Noisy composition.** To simulate processing through a DNN, we add Gaussian noise to the composed vector. In particular,

$$\hat{\boldsymbol{c}} = \boldsymbol{c} + \boldsymbol{e} \tag{4}$$

with $\boldsymbol{e} \in \mathbb{R}^d, \mathbf{e}_i \sim \mathcal{N}(0, \sigma^2)$ i.i.d.. $\sigma^2 = \dfrac{\|\boldsymbol{c}\|_2^2}{d \times \mathrm{SNR}}$ is chosen to maintain a constant SNR at all embedding sizes. Then, decomposition is performed on $\hat{\boldsymbol{c}}$ instead of $\boldsymbol{c}$. In practice, we add centered Gaussian noise to reach an SNR of $0\,\mathrm{dB}$ and $-2\,\mathrm{dB}$.

**Multi-level composition.** We create a hierarchical structure by performing multiple compositions and decompositions at once. To do so, we generate $L$ codebooks $K^{l \in [1,...,L]}$ — one for each level — but still only one $V$. The indices are now divided into levels $I_i^l$, with the same overall number of elements as before. Each composition shares the same structure as above, with an added weighting factor $w_l$, and the final multi-level composition is computed by summation

$$\boldsymbol{c} = \sum_{l=1}^{L} w_l \sum_{i=1}^{N/L} K_i^l \otimes V_{I_i^l} \tag{5}$$

The decomposition is performed analogously for each level

$$\hat{I}_i^l = \mathrm{argmax}_i(\mathrm{sim}(\boldsymbol{c} \oslash K^l, V)) \tag{6}$$

We use $L = 4$ levels and $N = 32$ vectors per level, always maintaining a total of 128. We use weight values $w = [1.3, 1.2, 1.1, 1.0]$ in the weighted multi-level case.

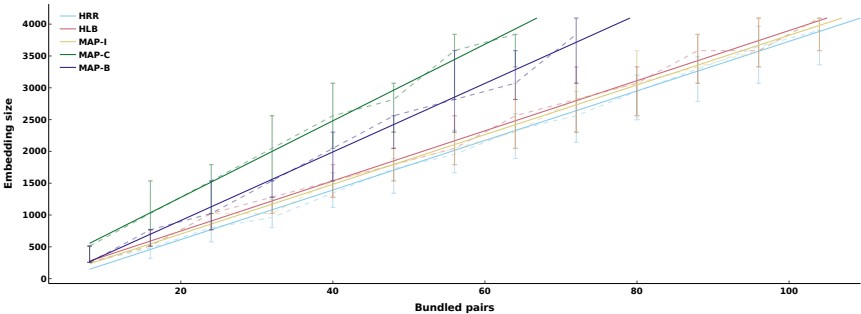

Figure 3: Retrieval effectiveness of all tested VSA families, reported as the minimum number of dimensions required to reach 99% accuracy. The solid lines are the best linear fit, while the dashed lines are the corresponding data. The error bars are the required dimensions with one standard deviation from the mean accuracy.

The hardware used is a single NVIDIA A100 with 80 GB of VRAM. The software stack includes PyTorch 2.4.0 and FlashFFTConv (commit b877102). All runs are performed on GPU. Full results for each VSA model can be found in Appendix B.

## 4. Results

### 4.1. Runtime complexity

As training state-of-the-art DNNs can reach costs of millions of dollars (31), we must ensure that VSAs are as computationally effective as possible. We consider here three implementations of HRR, which uses circular convolutions, and HLB, which is purely linear (MAP is also linear, hence it has the same runtime complexity). Figure 2(a) shows that a naïve implementation of HRR using PyTorch's *Conv1D* module incurs a quadratic cost, making it unsuitable to larger implementations. Using a custom implementation with PyTorch's *fft* library reduces cost to loglinear, alleviating this bottleneck. Nonetheless, we can further increase the throughput by leveraging GPUs, and specifically custom kernels built for purpose such as FlashFFTConv (8). With this Flash implementation we can even match HLB's linear runtime.

To achieve additional speedup, we also evaluate running in reduced precision. The naïve implementation of HRR and HLB can be used in both full and reduced (bfloat16) precision. However, FlashFFTConv highly recommends half or mixed precision usage, while the FFT implementation of HRR can only be run in full precision. Figure 2(a) indicates that there is no benefit in runtime to reduced precision. At the same time, memory usage does decrease.

Memory consumption is, in fact, one of the major bottlenecks of VSA due to the cosine similarity computation required for the similarity search. Specifically, PyTorch's cosine similarity function runs out of 80GB VRAM at approximately 3000 embedding size, and only compiling it allows the benchmark to run by avoiding the materialization of intermediate matrices. Figure 2(b) shows the considerable speedup obtained by bypassing the similarity search through a linear readout (see Section 4.2 for more information about the retrieval

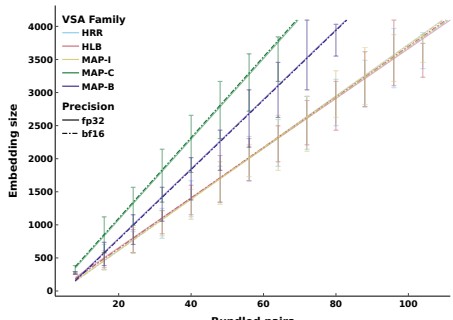

(a) Reduced precision does not affect retrieval.

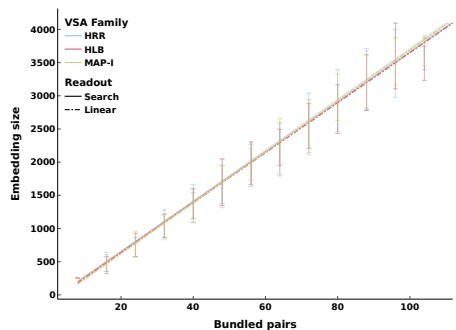

(b) Linear readout does not affect retrieval.

Figure 4: Retrieval accuracy of reduced precision and linear readout, reported as the minimum number of dimensions required to reach 99% accuracy.

performance). In particular, HRR using FFT sees an average speedup of 76%, and up to 80% at highest dimensionalities. HLB's speedup is even more significant, with an average of 85% and peaks of almost 90%. Removing this bottleneck also creates a clearer distance between HLB and HRR. Using regular FFT convolutions, HLB is between 70% to 30% faster than HRR, while with FlashFFT HLB reduces its advantage to 15% at higher dimensions.

## 4.2. Retrieval accuracy

We now evaluate all the considered VSA families in terms of their memory capacity. For each number of bundled pairs $N$, we compute the required size to reach 99% retrieval accuracy using cosine similarity. Figure 3 shows that there is no notable difference in retrieval accuracy between HRR, HLB, and MAP-I, while the other two variants of MAP are clearly less performant.

Next, we evaluate the effect of reduced precision. We have seen in Section 4.1 that bf16 has no effect on VSA runtime, but it is often used to train large DNNs efficiently and without running out of memory. Figure 4(a) shows that bf16 has no effect on the performance of any VSA family. Incidentally, we also do not find any improved stability for HLB when compared to MAP-I (4). Finally, we have previously observed that replacing the similarity search with a linear readout results in notable speedup. Here, we show in Figure 4(b) that this choice does not affect the retrieval performance of either HRR, HLB, or MAP-I. Moreover, we have found that the linear readout should be initialized with the transpose of the embedding, not the pseudoinverse (see Appendix A).

## 4.3. Noisy retrieval

Until now we have tested the composition and retrieval performance of VSA in an ideal scenario, where the composite vector is immediately decomposed into its components. However, integrating VSA into DNNs often requires further processing of the composite vector inside the DNN for later retrieval.

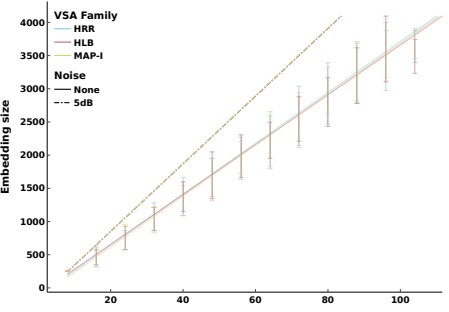

(a) VSA reconstruction is already affected at 5dB noise level.

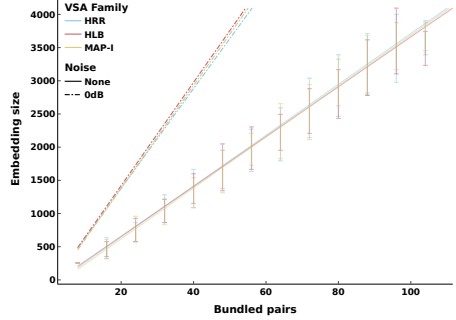

(b) At 0dB of noise, the relative performance of the VSA families does not change.

Figure 5: Retrieval accuracy of different noise levels, reported as the minimum number of dimensions required to reach 99% accuracy.

We model this scenario by injecting noise (see Section 3) into the composite vector, and later retrieving its components. Fig. 5 shows that HLB is again equivalent to HRR for this use-case. Increasing the noise from an SNR of 5dB (Fig. 5(a) to 0dB (Fig. 5(b)) affects the retrieval accuracy, it does so at the same rate for both. Once more, replacing similarity search with a linear readout does not compromise the accuracy (see Appendix A).

### 4.4. Multi-level composition

Finally, we aim to increase the practicality of VSA by showing it is capable of granular and hierarchical multi-level composition. In its most immediate form, multi-level composition only consists in binding the same symbol with two, instead of one, keys. Fig 6(b) shows that all levels are retrieved with the same accuracy. Moreover, the overall accuracy is the same as single level composition, given the same number of vectors to retrieve (see Section 3).

In many instances, not all internal representations share the same importance to the performance of the model. We achieve this effect by weighting the bound vectors differently during the bundling step. Figure 6(a) indicates that this scheme works, increasing the retrieval accuracy of the higher levels at the expense of the lower ones. Additionally, it does not carry any penalty to the overall reconstruction, which remains the same as before. Therefore, VSA can be successfully used for hierarchical multi-level composition without loss of performance.

## 5. Conclusion

We have analyzed the performance and effectiveness of various VSA models and implementations in the context of their potential use as composition and retrieval methods inside DNNs. First, we found that the computational complexity varies significantly across implementations of HRR, with naïve implementations being too cumbersome to be effectively

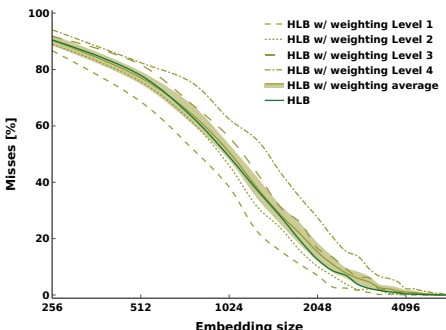 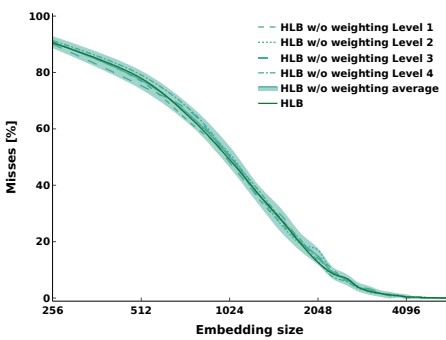

($a$) When the levels are weighted differently, the retrieval accuracy varies across levels.

($b$) When each level has the same weight, the retrieval accuracy is no different to having no levels.

Figure 6: Retrieval accuracy of HLB in a multi-level scenario. 32 vectors are assigned to each of 4 levels, and all vectors and levels are compressed into a single representation.

implemented in large DNNs. Indeed, most libraries adopt FFT convolutions (13; 1). Specialized GPU FFT implementations (8) further drive runtime down, close to linear VSAs such as HLB. Next, we found that the use of similarity search is a significant memory and speed bottleneck. Specifically, materialization of intermediate matrices quickly exhausts resources. Removing similarity search leads to a runtime improvement of 76% for HRR and 85% for HLB. At the same time, we found that this replacement does not affect retrieval accuracy. More advanced decoding methods could also be used, e.g., MMSE readout (6) or iterative algorithms (14; 26), after careful performance evaluation. We also tested the resistance of VSAs to noise, and found that while all considered models are affected, their relative accuracy does not change in this scenario. Finally, we deploy a practical solution to achieve hierarchical multi-level composition using VSA. Using simple weighting, we can affect the retrieval rates of the various levels, without compromising overall accuracy.

We have performed a thorough evaluation of HRR, HLB, and MAP, and provided some practical lessons to simplify the implementation of VSAs. Still, the usage of a specific model must be evaluated thoroughly on a case-by-case basis to avoid unforeseen bottlenecks and pitfalls. Moreover, the use-case we have provided is only one of possibly many integrations of VSA inside neural networks, especially in the context of NVSA. Therefore, while the runtime and memory performance results are generally valid for the VSA families considered, their contribution to the overall computational burden must be evaluated for the specific architecture. Finally, while we have selected what we believe are the most popular and effective choices, many more VSA models exist that could be successfully implemented inside DNNs.

## Acknowledgments

This work is supported by the Swiss National Science foundation (SNF), grant no. 200800.

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

## Appendix A. Additional lessons

### A.1. Linear readout with pseudoinverse initialization

We also perform the linear readout by changing the initialization from the transpose of the value matrix to its pseudoinverse. This solution performs notably worse than the more traditional similarity search as can be seen in Figure 7.

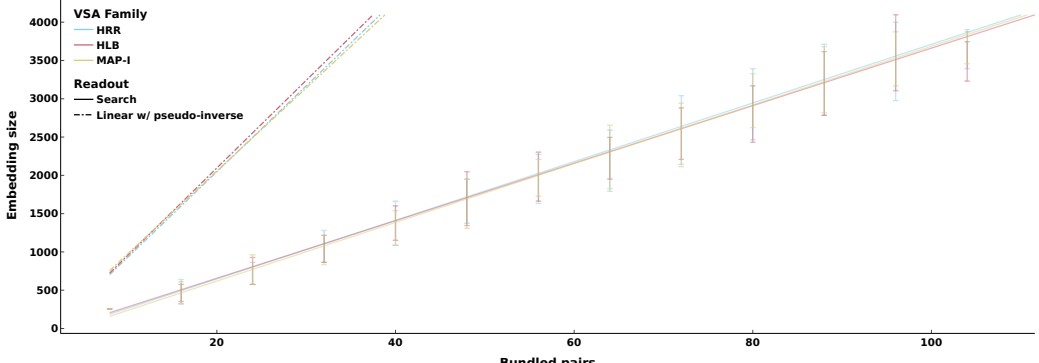

Figure 7: Retrieval effectiveness of the linear readout initialized with the pseudoinverse, reported as the minimum number of dimensions required to reach 99% accuracy. The solid lines are the best linear fit, while the dashed lines are the corresponding data. The error bars are the required dimensions with one standard deviation from the mean accuracy.

### A.2. Noisy linear readout

We compare the effect of noise on the linear readout, to evaluate any difference in robustness between linear readout and cosine similarity search. Figure 8 shows that linear readout also does not affect accuracy under noisy conditions.

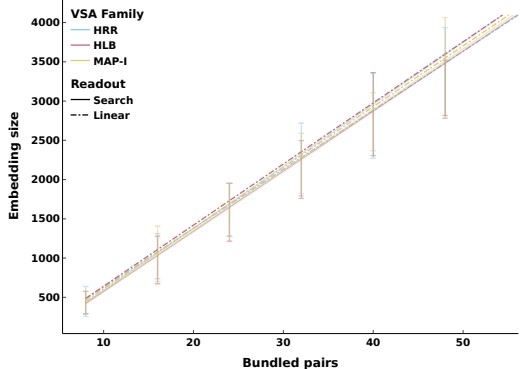

(a) Linear readout does not affect retrieval with 0dB noise.

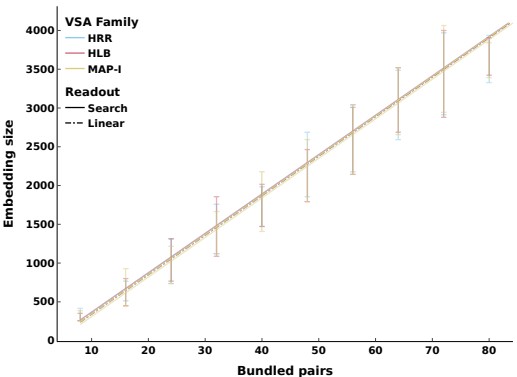

(b) Linear readout does not affect retrieval with 5dB noise.

Figure 8: Retrieval accuracy of noisy linear readout, reported as the minimum number of dimensions required to reach 99% accuracy. Solid lines are the best linear fit of the baseline, while dot-dashed lines are the best linear fit of the specific setting. The error bars are the required dimensions with one standard deviation from the mean accuracy.

## Appendix B. Full results

We provide in this Appendix the full results for each VSA family. In particular, we provide the accuracy across all embedding sizes and number of bundled pairs.

### B.1. HRR

Figure 9 shows the accuracy of HRR in all the tasks.

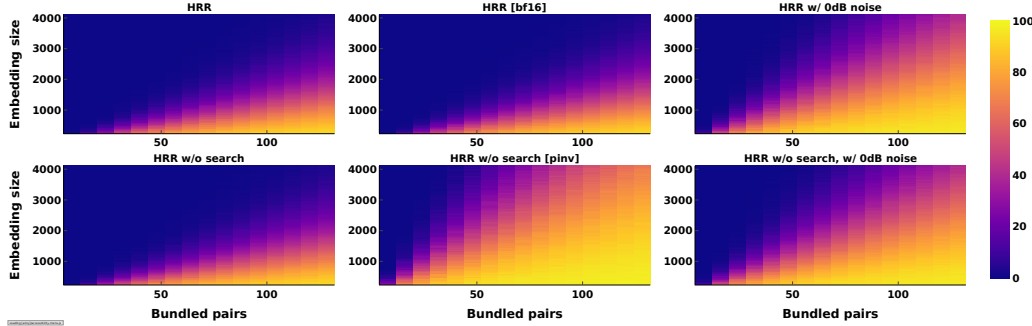

Figure 9: Retrieval effectiveness of HRR across all tested scenarios.

### B.2. HLB

Figure 10 shows the accuracy of HLB in all the tasks.

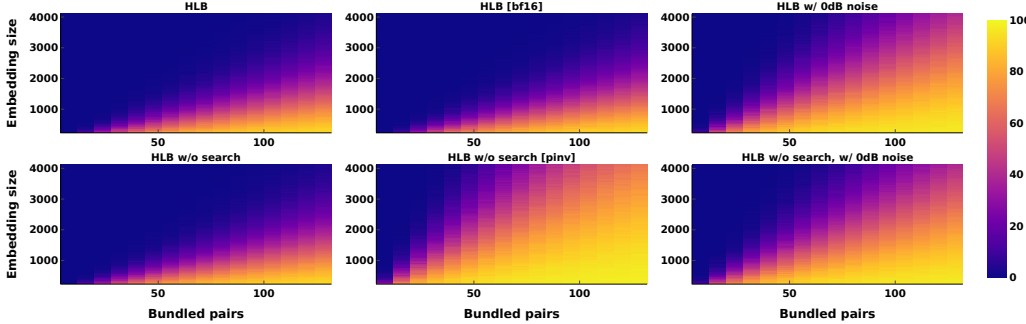

Figure 10: Retrieval effectiveness of HLB across all tested scenarios.

### B.3. MAP-I

Figure 11 shows the accuracy of MAP-I in all the tasks.

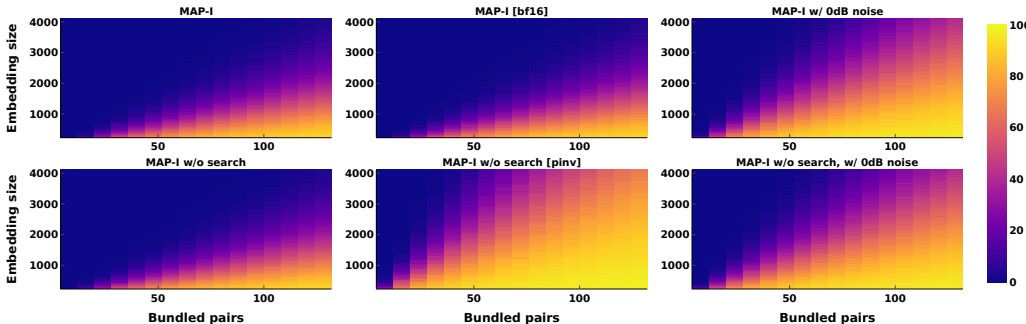

Figure 11: Retrieval effectiveness of MAP-I across all tested scenarios.

### B.4. MAP-C

Figure 12 shows the accuracy of MAP-C in the fp32 and bf16 tasks only. Given the poor performance of this model relative to the others, we do not investigate the other scenarios.

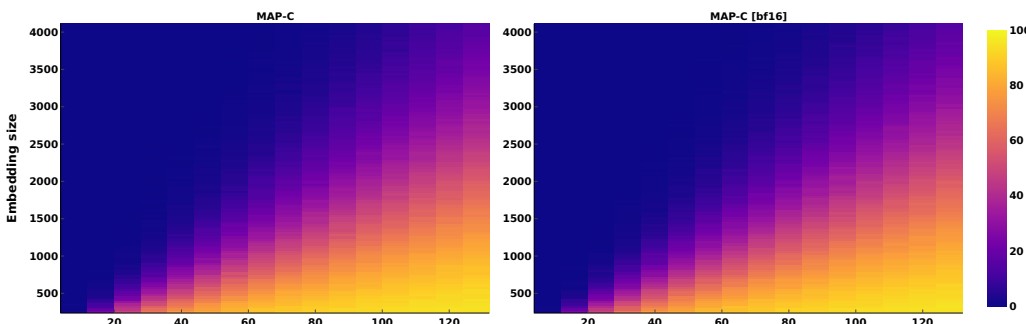

Figure 12: Retrieval effectiveness of MAP-C with fp32 and bf16 precision.

### B.5. MAP-B

Figure 13 shows the accuracy of MAP-B in the fp32 and bf16 tasks only. Given the poor performance of this model relative to the others, we do not investigate the other scenarios.

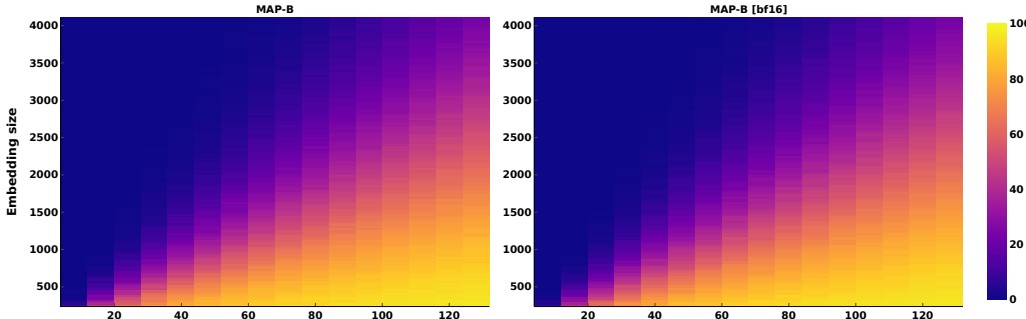

Figure 13: Retrieval effectiveness of MAP-B across all tested scenarios. Given the poor performance of this model relative to the others, we do not investigate the other scenarios.

