# OpenReview forum: "Practical Lessons on Vector-Symbolic Architectures in Deep Learning-Inspired Environments"
_nesyconf.org/NeSy/2025/Conference_Phase_2 — NeSy 2025 - Phase 2 Poster_

### Official Review · Reviewer_EFNm · 2025-07-01
**Good empirical contributions but research questions are not clearly stated**

**Rating:** 4
**Confidence:** 3

**Review:**

- The paper provides empirical improvements in Vector-Symbolic
Architectures. The paper is easy to follow, but I believe it lacks
several key components.

- The paper contains several practical contributions that can be of
interest of practitioners. However, I think these contributions would
be more suitable for a workshop instead of a conference. The reason is
that it is hard to find what research question(s) are addressed in the
paper. Instead, the paper focuses on several practical improvements on
VSA architectures. If authors intend to strengthen the paper towards a
conference or journal version I would suggest answering these
questions:

  - What specific research questions are being investigated?
  - Why do these questions would be of interest of research community?
  - How does your methodology help to address the research questions?
  - What are the limitations of your methods?


- The paper highlights time efficiency as the core problem of existing
  VSA operations. I would suggest formulating research question in
  that direction. Thus, Section 3 (which contains your contributions)
  would be the 'Methods' section. So far the jump from 'Background' to
  'Experimental Setup' feels strange and seems like there is no any
  new methodological contribution but just a bunch of experiments.

**Anonymity:**

Remain anonymous

---

### Official Review · Reviewer_Jpji · 2025-07-08
**Empirically investigating VSA for Deep Learning**

**Rating:** 6
**Confidence:** 2

**Review:**

Summary:
This paper explores how a class of symbolic reasoning tools called Vector-Symbolic Architectures (VSAs) can be used efficiently alongside modern deep learning systems. The authors provide four practical lessons, backed by  experiments, on how to make VSA models more compatible with neural networks. Their focus is on optimizing speed, memory usage, and maintaining accuracy in symbolic retrieval tasks.

Strengths:
- Clear Practical Focus: The paper offers concrete advice on improving the computational efficiency of VSA models when integrated with neural networks.
- Strong Experimental Backing: Extensive experiments compare different VSA variants (HRR, MAP, HLB) under varying conditions (e.g., noise, reduced precision).
- Relevance to Real-World Systems: Implementation concerns like GPU compatibility and memory bottlenecks are directly addressed.
- Accessibility: The structure and visualizations make the findings relatively easy to understand, for non-specialists (like myself).

Weaknesses:
- Limited Theoretical Discussion: The work emphasizes engineering improvements, with little discussion of theoretical trade-offs or model limitations.

Decision:
This is a potentially useful and practical paper. While it doesn’t introduce new models or theoretical insights, it offers empirical guidance for using VSA techniques efficiently in modern architectures.

**Anonymity:**

Remain anonymous